# Learning Self-Correctable Policies and Value Functions from Demonstrations with Negative Sampling

**Yuping Luo**
Princeton University
yupingl@cs.princeton.edu

**Huazhe Xu**
University of California, Berkeley
huazhe_xu@eecs.berkeley.edu

**Tengyu Ma**
Stanford University
tengyuma@stanford.edu

## Abstract

Imitation learning, followed by reinforcement learning algorithms, is a promising paradigm to solve complex control tasks sample-efficiently. However, learning from demonstrations often suffers from the covariate shift problem, which results in cascading errors of the learned policy. We introduce a notion of conservatively-extrapolated value functions, which provably lead to policies with self-correction. We design an algorithm Value Iteration with Negative Sampling (VINS) that practically learns such value functions with conservative extrapolation. We show that VINS can correct mistakes of the behavioral cloning policy on simulated robotics benchmark tasks. We also propose the algorithm of using VINS to initialize a reinforcement learning algorithm, which is shown to outperform prior works in sample efficiency.

## 1 Introduction

Reinforcement learning (RL) algorithms, especially with sparse rewards, often require a large amount of trial-and-errors. Imitation learning from a small number of demonstrations followed by RL fine-tuning is a promising paradigm to improve the sample efficiency (Rajeswaran et al., 2017; Večerík et al., 2017; Hester et al., 2018; Nair et al., 2018; Gao et al., 2018).

The key technical challenge of learning from demonstrations is the covariate shift: the distribution of the states visited by the demonstrations often has a low-dimensional support; however, knowledge learned from this distribution may not necessarily transfer to other distributions of interests. This phenomenon applies to both learning the policy and the value function. The policy learned from behavioral cloning has compounding errors after we execute the policy for multiple steps and reach unseen states (Bagnell, 2015; Ross & Bagnell, 2010). The value function learned from the demonstrations can also extrapolate falsely to unseen states. See Figure 1a for an illustration of the false extrapolation in a toy environment.

We develop an algorithm that learns a value function that extrapolates to unseen states more conservatively, as an approach to attack the optimistic extrapolation problem (Fujimoto et al., 2018a). Consider a state $s$ in the demonstration and its nearby state $\tilde{s}$ that is not in the demonstration. The key intuition is that $\tilde{s}$ should have a lower value than $s$, because otherwise $\tilde{s}$ likely should have been visited by the demonstrations in the first place. If a value function has this property for most of the pair $(s, \tilde{s})$ of this type, the corresponding policy will tend to correct its errors by driving back to the demonstration states because the demonstration states have locally higher values. We formalize the intuition in Section 4 by defining the so-called conservatively-extrapolated value function, which is guaranteed to induce a policy that stays close to the demonstrations states (Theorem 4.4).

In Section 5, we design a practical algorithm for learning the conservatively-extrapolated value function by a negative sampling technique inspired by work on learning embeddings Mikolov et al. (2013); Gutmann & Hyvärinen (2012). We also learn a dynamical model by standard supervised learning so that we compute actions by maximizing the values of the predicted next states. This algorithm does not use any additional environment interactions, and we show that it empirically helps correct errors of the behavioral cloning policy.

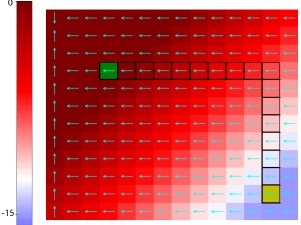 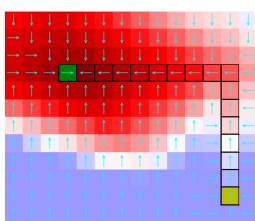

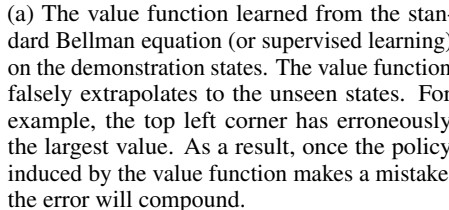

(a) The value function learned from the standard Bellman equation (or supervised learning) on the demonstration states. The value function falsely extrapolates to the unseen states. For example, the top left corner has erroneously the largest value. As a result, once the policy induced by the value function makes a mistake, the error will compound.

(b) The conservatively-extrapolated value function (defined in equation (4.2)) learned with negative sampling (VINS, Algorithm 2 in Section 5). The values at unseen states tend to be lower than their nearby states in the demonstrations, and therefore the corresponding policy tend to correct itself towards the demonstration trajectories.

Figure 1: A toy environment where the agent aims to walk from a starting state (the yellow entry) to a goal state (the green entry). The reward is sparse: $R(s, a) = -1$ unless $s$ is at the goal (which is also the terminal state.) The colors of the entries show the learned value functions. Entries in black edges are states in demonstrations. The cyan arrows show the best actions according to the value functions.

When additional environment interactions are available, we use the learned value function and the dynamical model to initialize an RL algorithm. This approach relieves the inefficiency in the prior work (Hester et al., 2018; Nair et al., 2018; Rajeswaran et al., 2017) that the randomly-initialized $Q$ functions require a significant amount of time and samples to be warmed up, even though the initial policy already has a non-trivial success rate. Empirically, the proposed algorithm outperforms the prior work in the number of environment interactions needed to achieve near-optimal success rate.

In summary, our main contributions are: 1) we formalize the notion of values functions with conservative extrapolation which are proved to induce policies that stay close to demonstration states and achieve near-optimal performances, 2) we propose the algorithm Value Iteration with Negative Sampling (VINS) that outperforms behavioral cloning on three simulated robotics benchmark tasks with sparse rewards, and 3) we show that initializing an RL algorithm from VINS outperforms prior work in sample efficiency on the same set of benchmark tasks.

## 2 RELATED WORK

**Imitation learning.** Imitation learning is commonly adopted as a standard approach in robotics (Pomerleau, 1989; Schaal, 1997; Argall et al., 2009; Osa et al., 2017; Ye & Alterovitz, 2017; Aleotti & Caselli, 2006; Lawitzky et al., 2012; Torabi et al., 2018; Le et al., 2017; 2018) and many other areas such as playing games (Mnih et al., 2013). Behavioral cloning (Bain & Sommut, 1999) is one of the underlying central approaches. See Osa et al. (2018) for a thorough survey and more references therein. If we are allowed to access an expert policy (instead of trajectories) or an approximate value function, in the training time or in the phase of collecting demonstrations, then, stronger algorithms can be designed, such as DAgger (Ross et al., 2011), AggreVaTe (Ross & Bagnell, 2014), AggreVaTeD (Sun et al., 2017), DART (Laskey et al., 2017), THOR Sun et al. (2018a). Our setting is that we have only clean demonstrations trajectories and a sparse reward (but we still hope to learn the self-correctable policy.)

Ho & Ermon (2016); Wang et al. (2017); Schroecker et al. (2018) successfully combine generative models in the setting where a large amount of environment interaction without rewards are allowed. The sample efficiency of (Ho & Ermon, 2016) has been improved in various ways, including maximum mean discrepancy minimization (Kim & Park, 2018), a Bayesian formulation of GAIL (Jeon et al., 2018), using an off-policy RL algorithm and solving reward bias problem (Kostrikov et al., 2018), and bypassing the learning of reward function (Sasaki et al., 2018). By contrast, we would like to minimize the amount of environment interactions needed, but are allowed to access a sparse reward. The work (Schroecker & Isbell, 2017) also aims to learn policies that can stay close to the demonstration sets, but through a quite different approach of estimating the true MAP estimate of the

policy. The algorithm also requires environment interactions, whereas one of our main goals is to improve upon behavioral cloning without any environment interactions.

Inverse reinforcement learning (e.g., see (Abbeel & Ng, 2004; Ng et al., 2000; Ziebart et al., 2008; Finn et al., 2016a;b; Fu et al., 2017)) is another important and successful line of ideas for imitation learning. It relates to our approach in the sense that it aims to learn a reward function that the expert is optimizing. In contrast, we construct a model to learn the value function (of the trivial sparse reward $R(s, a) = -1$), rather than the reward function. Some of these works (e.g., (Finn et al., 2016a;b; Fu et al., 2017)) use techniques that are reminiscent of negative sampling or contrastive learning, although unlike our methods, they use "negative samples" that are sampled from the environments.

**Leveraging demonstrations for sample-efficient reinforcement learning.** Demonstrations have been widely used to improve the efficiency of RL (Kim et al., 2013; Chemali & Lazaric, 2015; Piot et al., 2014; Sasaki et al., 2018), and a common paradigm for continuous state and action space is to initialize with RL algorithms with a good policy or $Q$ function (Rajeswaran et al., 2017; Nair et al., 2018; Večerík et al., 2017; Hester et al., 2018; Gao et al., 2018). We experimentally compare with the previous state-of-the-art algorithm in Nair et al. (2018) on the same type of tasks. Gao et al. (2018) has introduced soft version of actor-critic to tackle the false extrapolation of $Q$ in the argument of $a$ when the action space is discrete. In contrast, we deal with the extrapolation of the states in a continuous state and action space.

**Model-based reinforcement learning.** Even though we will learn a dynamical model in our algorithms, we do not use it to generate fictitious samples for planning. Instead, the learned dynamics are only used in combination with the value function to get a $Q$ function. Therefore, we do not consider our algorithm as model-based techniques. We refer to (Kurutach et al., 2018; Clavera et al., 2018; Sun et al., 2018b; Chua et al., 2018; Sanchez-Gonzalez et al., 2018; Pascanu et al., 2017; Khansari-Zadeh & Billard, 2011; Luo et al., 2018) and the reference therein for recent work on model-based RL.

**Off-policy reinforcement learning** There is a large body of prior works in the domain of off-policy RL, including extensions of policy gradient (Gu et al., 2016; Degris et al., 2012; Wang et al., 2016) or Q-learning (Watkins & Dayan, 1992; Haarnoja et al., 2018; Munos et al., 2016). Fujimoto et al. (2018a) propose to solve off-policy reinforcement learning by constraining the action space, and Fujimoto et al. (2018c) use double Q-learning (Van Hasselt et al., 2016) to alleviate the optimistic extrapolation issue. In contrast, our method adjusts the erroneously extrapolated value function by explicitly penalizing the unseen states (which is customized to the particular demonstration off-policy data). For most of the off-policy methods, their convergence are based on the assumption of visiting each state-action pair sufficiently many times. In the learning from demonstration setting, the demonstrations states are highly biased or structured; thus off-policy method may not be able to learn much from the demonstrations.

## 3    PROBLEM SETUP AND CHALLENGES

We consider a setting with a deterministic MDP with continuous state and action space, and sparse rewards. Let $\mathcal{S} = \mathbb{R}^d$ be the state space and $\mathcal{A} = \mathbb{R}^k$ be the action space, and let $M^\star : \mathbb{R}^d \times \mathbb{R}^k \to \mathbb{R}^d$ be the deterministic dynamics. At test time, a random initial state $s_0$ is generated from some distribution $D_{s_0}$. We assume $D_{s_0}$ has a low-dimensional bounded support because typically initial states have special structures. We aim to find a policy $\pi$ such that executing $\pi$ from state $s_0$ will lead to a set of goal states $\mathcal{G}$. All the goal states are terminal states, and we run the policy for at most $T$ steps if none of the goal states is reached.

Let $\tau = (s_0, a_1, s_1, \dots, )$ be the trajectory obtained by executing a deterministic policy $\pi$ from $s_0$, where $a_t = \pi(s_t)$, and $s_{t+1} = M^\star(s_t, a_t)$. The success rate of the policy $\pi$ is defined as

$$\text{succ}(\pi) = \mathbb{E}\left[\mathbb{1}\{\exists t \le T, s_t \in \mathcal{G}\}\right] \tag{3.1}$$

where the expectation is taken over the randomness of $s_0$. Note that the problem comes with a natural sparse reward: $R(s, a) = -1$ for every $s$ and $a$. This will encourage reaching the goal with as small number of steps as possible: the total payoff of a trajectory is equal to negative the number of steps if the trajectory succeeds, or $-T$ otherwise.

Let $\pi_\mathrm{e}$ be an expert policy [1] from which a set of $n$ demonstrations are sampled. Concretely, $n$ independent initial states $\{s_0^{(i)}\}_{i=1}^n$ from $D_{s_0}$ are generated, and the expert executes $\pi_\mathrm{e}$ to collect a set of $n$ trajectories $\{\tau^{(i)}\}_{i=1}^n$. We only have the access to the trajectories but not the expert policy itself.

We will design algorithms for two different settings:

**Imitation learning without environment interactions:** The goal is to learn a policy $\pi$ from the demonstration trajectories $\{\tau^{(i)}\}_{i=1}^n$ without having any additional interactions with the environment.

**Leveraging demonstrations in reinforcement learning:** Here, in addition to the demonstrations, we can also interact with the environment (by sampling $s_0 \sim D_{s_0}$ and executing a policy) and observe if the trajectory reaches the goal. We aim is to minimize the amount of environment interactions by efficiently leveraging the demonstrations.

Let $\mathcal{U}$ be the set of states that can be visited by the demonstration policy from a random state $s_0$ with positive probability. Throughout this paper, *we consider the situation where the set $\mathcal{U}$ is only a small subset or a low-dimensional manifold of the entire state space.* This is typical for continuous state space control problems in robotics, because the expert policy may only visit a very special kind of states that are the most efficient for reaching the goal. For example, in the toy example in Figure 1, the set $\mathcal{U}$ only contains those entries with black edges.[2]

To put our theoretical motivation in Section 4 into context, next we summarize a few challenges of imitation learning that are particularly caused by that $\mathcal{U}$ is only a small subset of the state space.

**Cascading errors for behavioral cloning.** As pointed out by Bagnell (2015); Ross & Bagnell (2010), the errors of the policy can compound into a long sequence of mistakes and in the worst case cascade quadratically in the number of time steps $T$. From a statistical point of view, the fundamental issue is that the distribution of the states that a learned policy may encounter is different from the demonstration state distribution. Concretely, the behavioral cloning $\pi_\mathrm{BC}$ performs well on the states in $\mathcal{U}$ but not on those states far away from $\mathcal{U}$. However, small errors of the learned policy can drive the state to leave $\mathcal{U}$, and then the errors compound as we move further and further away from $\mathcal{U}$. As shown in Section 4, our key idea is to design policies that correct themselves to stay close to the set $\mathcal{U}$.

**Degeneracy in learning value or $Q$ functions from only demonstrations.** When $\mathcal{U}$ is a small subset or a low-dimensional manifold of the state space, off-policy evaluation of $V^{\pi_\mathrm{e}}$ and $Q^{\pi_\mathrm{e}}$ is fundamentally problematic in the following sense. The expert policy $\pi_\mathrm{e}$ is not uniquely defined outside $\mathcal{U}$ because any arbitrary extension of $\pi_\mathrm{e}$ outside $\mathcal{U}$ would not affect the performance of the expert policy (because those states outside $\mathcal{U}$ will never be visited by $\pi_\mathrm{e}$ from $s_0 \sim D_{s_0}$). As a result, the value function $V^{\pi_\mathrm{e}}$ and $Q^{\pi_\mathrm{e}}$ is not uniquely defined outside $\mathcal{U}$. In Section 4, we will propose a conservative extrapolation of the value function that encourages the policy to stay close to $\mathcal{U}$. Fitting $Q^{\pi_\mathrm{e}}$ is in fact even more problematic. We refer to Section A for detailed discussions and why our approach can alleviate the problem.

**Success and challenges of initializing RL with imitation learning.** A successful paradigm for sample-efficient RL is to initialize the RL policy by some coarse imitation learning algorithm such as BC (Rajeswaran et al., 2017; Večerík et al., 2017; Hester et al., 2018; Nair et al., 2018; Gao et al., 2018). However, the authors suspect that the method can still be improved, because the value function or the $Q$ function are only randomly initialized so that many samples are burned to warm them up. As alluded before and shown in Section 4, we will propose a way to learn a value function from the demonstrations so that the following RL algorithm can be initialized by a policy, value function, and $Q$ function (which is a composition of value and dynamical model) and thus converge faster.

## 4 THEORETICAL MOTIVATIONS

In this section, we formalize our key intuition that the ideal extrapolation of the value function $V^{\pi_\mathrm{e}}$ should be that the values should decrease as we get further and further from the demonstrations. Recall that we use $\mathcal{U}$ to denote the set of states reachable by the expert policy from any initial state $s_0$

---

[1] In this work, we only consider deterministic expert policies.

[2] One may imagine that $\mathcal{U}$ can be a more diverse set if the demonstrations are more diverse, but an expert will not visit entries on the top or bottom few rows, because they are not on any optimal routes to the goal state.

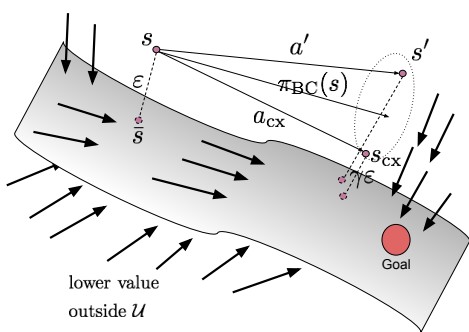

Figure 2: Illustration of the correction effect. A conservatively-extrapolated value function $V$, as shown in the figure, has lower values further away from $\mathcal{U}$, and therefore the gradients of $V$ point towards $\mathcal{U}$. With such a value function, suppose we are at state $s$ which is $\varepsilon$-close to $\mathcal{U}$. The locally-correctable assumption of the dynamics assumes the existence of $a_{\mathrm{cx}}$ that will drive us to state $s_{\mathrm{cx}}$ that is closer to $\mathcal{U}$ than $s$. Since $s_{\mathrm{cx}}$ has a relatively higher value compared to other possible future states that are further away from $\mathcal{U}$ (e.g., $s'$ shown in the figure), $s_{\mathrm{cx}}$ will be preferred by the optimization (4.3). In other words, if an action $a$ leads to state $s$ with large distance to $\mathcal{U}$, the action won't be picked by (4.3) because it cannot beat $a_{\mathrm{cx}}$.

drawn with positive probability from $D_{s_0}$. [3] We use $\|\cdot\|$ to denote a norm in Euclidean space $\mathbb{R}^d$. Let $\Pi_{\mathcal{U}}(s)$ be the projection of $s \in \mathbb{R}^d$ to a set $\mathcal{U} \subset \mathbb{R}^d$ (according to the norm $\|\cdot\|$) [4].

We introduce the notion of value functions with conservative extrapolation which matches $V^{\pi_e}$ on the demonstration states $\mathcal{U}$ and has smaller values outside $\mathcal{U}$. As formally defined in equation (4.1) and (4.2) in Alg. 1, we extrapolate $V^{\pi_e}$ in a way that the value at $s \notin \mathcal{U}$ is decided by the value of its nearest neighbor in $\mathcal{U}$ (that is $V^{\pi_e}(\Pi_{\mathcal{U}}(s))$), and its distance to the nearest neighbor (that is, $\|s - \Pi_{\mathcal{U}}(s)\|$). We allow a $\delta_V > 0$ error because exact fitting inside or outside $\mathcal{U}$ would be impossible.

---

**Algorithm 1** Self-correctable policy induced from a value function with conservative extrapolation

**Require: conservatively-extrapolated values** $V$ satisfying

$$V(s) = V^{\pi_e}(s) \pm \delta_V, \qquad\qquad \text{if } s \in \mathcal{U} \qquad (4.1)$$
$$V(s) = V^{\pi_e}(\Pi_{\mathcal{U}}(s)) - \lambda\|s - \Pi_{\mathcal{U}}(s)\| \pm \delta_V \qquad\qquad \text{if } s \notin \mathcal{U} \qquad (4.2)$$

and a locally approximately correct dynamics $M$ and BC policy $\pi_{\mathrm{BC}}$ satisfying Assumption (4.1).

**Self-correctable policy $\pi$:**

$$\pi(s) \triangleq \underset{a:\|a-\pi_{\mathrm{BC}}(s)\|\leq\zeta}{\mathrm{argmax}} \; V(M(s,a)) \qquad (4.3)$$

---

Besides a conservatively-extrapolated value function $V$, our Alg. 1 relies on a learned dynamical model $M$ and a behavioral cloning policy $\pi_{\mathrm{BC}}$. With these, the policy returns the action with the maximum value of the predicted next state in around the action of the BC policy. In other words, the policy $\pi$ attempts to re-adjust the BC policy locally by maximizing the value of the next state.

Towards analyzing Alg. 1, we will make a few assumptions. We first assume that the BC policy is correct in the set $\mathcal{U}$, and the dynamical model $M$ is locally correct around the set $\mathcal{U}$ and the BC actions. Note that these are significantly weaker than assuming that the BC policy is globally correct (which is impossible to ensure) and that the model $M$ is globally correct.

**Assumption 4.1** (Local errors in learned dynamics and BC policy). We assume the BC policy $\pi_{\mathrm{BC}}$ makes at most $\delta_\pi$ error in $\mathcal{U}$: for all $s \in \mathcal{U}$, we have $\|\pi_{\mathrm{BC}}(s) - \pi_e(s)\| \leq \delta_\pi$. We also assume that the learned dynamics $M$ has $\delta_M$ error locally around $\mathcal{U}$ and the BC actions in the sense that for all $s$ that is $\varepsilon$-close to $\mathcal{U}$, and any action that is $\zeta$-close to $\pi_{\mathrm{BC}}(s)$, we have $\|M(s,a) - M^\star(s,a)\| \leq \delta_M$.

We make another crucial assumption on the stability/correctability of the true dynamics. The following assumption essentially says that if we are at a state that is near the demonstration set, then there exists an action that can drive us closer to the demonstration set. This assumption rules out certain dynamics that does not allow corrections even after the policy making a small error. For example, if

---

[3]Recall that we assume that $D_{s_0}$ has a low-dimensional support and thus typically $\mathcal{U}$ will also be a low-dimensional subset of the ambient space.

[4]Any tiebreaker can be used if there are multiple closest points.

a robot, unfortunately, falls off a cliff, then fundamentally it cannot recover itself — our algorithm cannot deal with such pathological situations.

**Assumption 4.2** (Locally-correctable dynamics). For some $\gamma \in (0,1)$ and $\varepsilon > 0, L_c > 0$, we assume that the dynamics $M^\star$ is $(\gamma, L_c, \varepsilon)$-locally-correctable w.r.t to the set $\mathcal{U}$ in the sense that for all $\varepsilon_0 \in (0, \varepsilon]$ and any tuple $(\bar{s}, \bar{a}, \bar{s}')$ satisfying $\bar{s}, \bar{s}' \in \mathcal{U}$ and $\bar{s}' = M^\star(\bar{s}, \bar{a})$, and any $\varepsilon_0$-perturbation $s$ of $\bar{s}$ (that is, $s \in N_{\varepsilon_0}(\bar{s})$), there exists an action $a_{\mathrm{cx}}$ that is $L_c \varepsilon_0$ close to $\bar{a}$, such that it makes a correction in the sense that the resulting state $s'$ is $\gamma \varepsilon_0$-close to the set $\mathcal{U}$: $s' = M^\star(s, a_{\mathrm{cx}}) \in N_{\gamma \varepsilon_0}(\mathcal{U})$. Here $N_\delta(K)$ denotes the set of points that are $\delta$-close to $K$.

Finally, we will assume the BC policy, the value function, and the dynamics are all Lipschitz in their arguments.[5] We also assume the projection operator to the set $\mathcal{U}$ is locally Lipschitz. These are regularity conditions that provide loose local extrapolation of these functions, and they are satisfied by parameterized neural networks that are used to model these functions.

**Assumption 4.3** (Lipschitz-ness of policy, value function, and dynamics). We assume that the policy $\pi_{\mathrm{BC}}$ is $L_\pi$-Lipschitz. That is, $\|\pi_{\mathrm{BC}}(s) - \pi_{\mathrm{BC}}(\tilde{s})\| \le L_\pi \|s - \tilde{s}\|$ for all $s, \tilde{s}$. We assume the value function $V^{\pi_{\mathrm{e}}}$ and the learned value function $V$ are $L_V$-Lipschitz, the model $M^\star$ is $L_{M,a}$-Lipschitz w.r.t to the action and $L_{M,s}$-Lipschitz w.r.t to the state $s$. We also assume that the set $\mathcal{U}$ has $L_\Pi$-Lipschitz projection locally: for all $s, \hat{s}$ that is $\varepsilon$-close to $\mathcal{U}$, $\|\Pi_{\mathcal{U}}(s) - \Pi_{\mathcal{U}}(\hat{s})\| \le L_\Pi \|s - \hat{s}\|$.

Under these assumptions, now we are ready to state our main theorem. It claims that 1) the induced policy $\pi$ in Alg. 1 stays close to the demonstration set and performs similarly to the expert policy $\pi_{\mathrm{e}}$, and 2) following the induced policy $\pi$, we will arrive at a state with a near-optimal value.

**Theorem 4.4.** *Suppose Assumption 4.1, 4.2, 4.3 hold with sufficiently small $\varepsilon > 0$ and errors $\delta_M, \delta_\pi, \delta_\pi > 0$ so that they satisfy $\zeta \ge L_c \varepsilon + \delta_\pi + L_\pi$. Let $\lambda$ be sufficiently large so that $\lambda \ge \frac{2 L_V L_\Pi L_M \zeta + 2 \delta_V + 2 L_V \delta_M}{(1-\gamma)\varepsilon}$. Then, the policy $\pi$ from equation (4.3) satisfies the following:*

1. *Starting from $s_0 \in \mathcal{U}$ and executing policy $\pi$ for $T_0 \le T$ steps, the resulting states $s_1, \ldots, s_{T_0}$ are all $\varepsilon$-close to the demonstrate states set $\mathcal{U}$.*

2. *In addition, suppose the expert policy makes at least $\rho$ improvement every step in the sense that for every $s \in \mathcal{U}$, either $V^{\pi_{\mathrm{e}}}(M^\star(s, \pi_{\mathrm{e}}(s))) \ge V^{\pi_{\mathrm{e}}}(s) + \rho$ or $M^\star(s, \pi_{\mathrm{e}}(s))$ reaches the goal.[6] Assume $\varepsilon$ and $\delta_M, \delta_V, \delta_\pi$ are small enough so that they satisfy $\rho \gtrsim \varepsilon + \delta_\pi$.*

   *Then, the policy $\pi$ will achieve a state $s_T$ with $T \le 2|V^{\pi_{\mathrm{e}}}(s_0)|/\rho$ steps which is $\varepsilon$-close to a state $\bar{s}_T$ with value at least $V^{\pi_{\mathrm{e}}}(s_T) \gtrsim -(\varepsilon + \delta_\pi)$.[7]*

The proof is deferred to Section B. The first bullet follows inductively invoking the following lemma which states that if the current state is $\varepsilon$-close to $\mathcal{U}$, then so is the next state. The proof of the Lemma is the most mathematically involved part of the paper and is deferred to the Section B. We demonstrate the key idea of the proof in Figure 2 and its caption.

**Lemma 4.5.** *In the setting of Theorem 4.4, suppose $s$ is $\varepsilon$-close to the demonstration states set $\mathcal{U}$. Suppose $\mathcal{U}$, and let $a = \pi(s)$ and $s' = M^\star(s, a)$. Then, $s'$ is also $\varepsilon$-close to the set $\mathcal{U}$.*

We effectively represent the $Q$ function by $V(M(s, a))$ in Alg. 1. We argue in Section A.1 that this helps address the degeneracy issue when there are random goal states (which is the case in our experiments.)

**Discussion: can we learn conservatively-extrapolated $Q$-function?** We remark that we do not expect a conservative-extrapolated $Q$-functions would be helpful. The fundamental idea here is to penalize the value of unseen states so that the policy can self-correct. However, to learn a $Q$ function that induces self-correctable policies, we should *encourage* unseen actions that can correct the trajectory, instead of penalize them just because they are not seen before. Therefore, it is crucial that the penalization is done on the unseen states (or $V$) but not the unseen actions (or $Q$).

---

[5]We note that technically when the reward function is $R(s, a) = -1$, the value function is not Lipschitz. This can be alleviated by considering a similar reward $R(s, a) = -\alpha - \beta\|a\|^2$ which does not require additional information.

[6]$\rho$ is 1 when the reward is always $-1$ before achieving the goal.

[7]Here $\gtrsim$ hides multiplicative constant factors depending on the Lipschitz parameters $L_{M,a}, L_{M,s}, L_\pi, L_V$.

---

**Algorithm 2** Value Iteration on Demonstrations with Negative Sampling (VINS)

---

1: $\mathcal{R} \leftarrow$ demonstration trajectories               $\triangleright$ No environment interaction will be used
2: Initialize value parameters $\bar{\phi} = \phi$ and model parameters $\theta$ randomly
3: **for** $i = 1, \ldots, T$ **do**
4:      sample mini-batch $\mathcal{B}$ of $N$ transitions $(s, a, r, s')$ from $\mathcal{R}$
5:      update $\phi$ to minimize $\mathcal{L}_{td}(\phi; \mathcal{B}) + \mathcal{L}_{ns}(\phi; \mathcal{B})$
6:      update $\theta$ to minimize loss $\mathcal{L}_{\mathrm{model}}(\theta; \mathcal{B})$
7:      update target network: $\bar{\phi} \leftarrow \bar{\phi} + \tau(\phi - \bar{\phi})$
8:
9: **function** POLICY($s$)
10:      Option 1: $a = \pi_{\mathrm{BC}}(s)$; Option 2: $a = 0$
11:      sample $k$ noises $\xi_1, \ldots, \xi_k$ from Uniform$[-1, 1]^m$      $\triangleright$ $m$ is the dimension of action space
12:      $i^* = \mathrm{argmax}_i V_\phi(M_\theta(s, a + \alpha\xi_i))$            $\triangleright$ $\alpha > 0$ is a hyper-parameter
13:      **return** $a + \alpha\xi_{i^*}$

---

## 5   MAIN APPROACH

**Learning value functions with negative sampling from demonstration trajectories.** As motivated in Section 4 by Algorithm 1 and Theorem 4.4, we first develop a practical method that can learn a value function with conservative extrapolation, without environment interaction. Let $V_\phi$ be a value function parameterized by $\phi$. Using the standard TD learning loss, we can ensure the value function to be accurate on the demonstration states $\mathcal{U}$ (i.e., to satisfy equation (4.1)). Let $\bar{\phi}$ be the target value function,[8] the TD learning loss is defined as $\mathcal{L}_{td}(\phi) = \mathbb{E}_{(s,a,s') \sim \rho^{\pi_e}} \left[ \left( r(s, a) + V_{\bar{\phi}}(s') - V_\phi(s) \right)^2 \right]$ where $r(s, a)$ is the (sparse) reward, $\bar{\phi}$ is the parameter of the target network, $\rho^{\pi_e}$ is the distribution of the states-action-states tuples of the demonstrations. The crux of the ideas in this paper is to use a negative sampling technique to enforce the value function to satisfy conservative extrapolation requirement (4.2). It would be infeasible to enforce condition (4.2) for every $s \notin \mathcal{U}$. Instead, we draw random "negative samples" $\tilde{s}$ from the neighborhood of $\mathcal{U}$, and enforce the condition (4.2). This is inspired by the negative sampling approach widely used in NLP for training word embeddings Mikolov et al. (2013); Gutmann & Hyvärinen (2012). Concretely, we draw a sample $s \sim \rho^{\pi_e}$, create a random perturbation of $s$ to get a point $\tilde{s} \notin \mathcal{U}$. and construct the following loss function:[9]

$$\mathcal{L}_{ns}(\phi) = \mathbb{E}_{s \sim \rho^{\pi_e}, \tilde{s} \sim \mathrm{perturb}(s)} \left( V_{\bar{\phi}}(s) - \lambda\|s - \tilde{s}\| - V_\phi(\tilde{s}) \right)^2,$$

The rationale of the loss function can be best seen in the situation when $\mathcal{U}$ is assumed to be a low-dimensional manifold in a high-dimensional state space. In this case, $\tilde{s}$ will be outside the manifold $\mathcal{U}$ with probability 1. Moreover, the random direction $\tilde{s} - s$ is likely to be almost orthogonal to the tangent space of the manifold $\mathcal{U}$, and thus $s$ is a reasonable approximation of the projection of $\tilde{s}$ back to the $\mathcal{U}$, and $\|s - \tilde{s}\|$ is an approximation of $\|\Pi_{\mathcal{U}}\tilde{s} - \tilde{s}\|$. If $\mathcal{U}$ is not a manifold but a small subset of the state space, these properties may still likely to hold for a good fraction of $s$.

We only attempt to enforce condition (4.2) for states near $\mathcal{U}$. This likely suffices because the induced policy is shown to always stay close to $\mathcal{U}$. Empirically, we perturb $s$ by adding a Gaussian noise. The loss function to learn $V_\phi$ is defined as $\mathcal{L}(\phi) = \mathcal{L}_{td}(\phi) + \mu\mathcal{L}_{ns}(\phi)$ for some constant $\mu > 0$. For a mini-batch $\mathcal{B}$ of data, we define the corresponding empirical loss by $\mathcal{L}(\phi; \mathcal{B})$ (similarly we define $\mathcal{L}_{td}(\phi; \mathcal{B})$ and $\mathcal{L}_{ns}(\phi; \mathcal{B})$). The concrete iterative learning algorithm is described in line 1-7 of Algorithm 2 (except line 6 is for learning the dynamical model, described below.)

**Learning the dynamical model.** We use standard supervised learning to train the model. We use $\ell_2$ norm as the loss for model parameters $\theta$ instead of the more commonly used MSE loss, following the success of (Luo et al., 2018): $\mathcal{L}_{\mathrm{model}}(\theta) = \mathbb{E}_{(s,a,s') \sim \rho^{\pi_e}} \left[ \|M_\theta(s, a) - s'\|_2 \right]$.

**Optimization for policy.** We don't maintain an explicit policy but use an induced policy from $V_\phi$ and $M_\theta$ by optimizing equation (4.3). A natural choice would be using projected gradient ascent to

---

[8]A target value function is widely used in RL to improve the stability of the training (Lillicrap et al., 2015; Mnih et al., 2015).

[9]With slight abuse of notation, we use $\rho^{\pi_e}$ to denote both the distribution of $(s, a, s')$ tuple and the distribution of $s$ of the expert trajectories.

optimize equation (4.3). It's also possible to use cross-entropy methods in (Kalashnikov et al., 2018) to optimize it. However, we found the random shooting suffices because the action space is relatively low-dimensional in our experiments. Moreover, the randomness introduced appears to reduce the overfitting of the model and value function slightly. As shown in line 10-13 of Alg. 2, we sample $k$ actions in the feasible set and choose the one with maximum $V_\phi(M_\theta(s, a))$.

**Value iteration with environment interaction.** As alluded before, when more environment interactions are allowed, we initialize an RL algorithm by the value function, dynamics learned from VINS. Given that we have $V$ and $M$ in hand, we alternate between fitted value iterations for updating the value function and supervised learning for updating the models. (See Algorithm 3 in Section C.) We do not use negative sampling here since the RL algorithms already collect bad trajectories automatically. We also do not hallucinate any goals as in HER (Andrychowicz et al., 2017).

## 6 EXPERIMENTS

**Environments.** We evaluate our algorithms in three simulated robotics environments[10] designed by (Plappert et al., 2018) based on OpenAI Gym (Brockman et al., 2016) and MuJoCo (Todorov et al., 2012): Reach, Pick-And-Place, and Push. A detailed description can be found in Section D.1.

**Demonstrations.** For each task, we use Hindsight Experience Replay (HER) (Andrychowicz et al., 2017) to train a policy until convergence. The policy rolls out to collect 100/200 successful trajectories as demonstrations except for Reach environment where 100 successful trajectories are sufficient for most of the algorithms to achieve optimal policy. We filtered out unsuccessful trajectories during data collection.

We consider two settings: imitation learning from only demonstrations data, and leveraging demonstration in RL with a *limited* amount of interactions. We compare our algorithm with Behavioral Cloning and multiple variants of our algorithms in the first setting. We compare with the previous state-of-the-art by Nair et al. (2018), and GAIL Ho & Ermon (2016) in the second setting. We do not compare with (Gao et al., 2018) because it cannot be applied to the case with continuous actions.

**Behavioral Cloning (Bain & Sommut, 1999).** Behavioral Cloning (BC) learns a mapping from a state to an action on demonstration data using supervised learning. We use MSE loss for predicting the actions.

**Nair *et al.*'18 (Nair et al., 2018).** The previous state-of-the-art algorithm from Nair et al. (2018) combines HER (Andrychowicz et al., 2017) with BC and a few techniques: 1) an additional replay buffer filled with demonstrations, 2) an additional behavioral cloning loss for the policy, 3) a $Q$-filter for non-optimal demonstrations, 4) resets to states in the demonstrations to deal with long horizon tasks. We note that reseting to an arbitrary state may not be realistic for real-world applications in robotics. In contrast, our algorithm does not require resetting to a demonstration state.

**GAIL (Ho & Ermon, 2016)** Generative Adversarial Imitation Learning (GAIL) imitates the expert by matching the state-action distribution with a GAN-like framework.

**HER (Andrychowicz et al., 2017)** Hindsight Experience Replay (HER) is the one of the best techniques that deal with sparse-reward environments with multiple *goals* and can be combined with any off-policy RL algorithm. The key idea is that HER extends the replay buffer by changing the goals. With reasonable chosen goals, the underlying off-policy RL algorihtm can receive more signals from the generated experience, making policy optimization more complete.

**DAC (Kostrikov et al., 2018)** Discriminator-Actor-Critic (DAC) is a sample-efficient imitation learning algorihtm built on the top of GAIL. It addresses the reward bias problem by adapting AIRL reward function and introducing an absorbing state. Furthermore, it replaces the underlying RL algorithm in GAIL by TD3 (Fujimoto et al., 2018b) to make it more sample efficient.

**VINS.** As described in Section 5, in the setting without environment interaction, we use Algorithm 2; otherwise we use it to initialize an RL algorithm (see Algorithm 3). We use neural networks to parameterize the value function and the dynamics model. The granularity of the HER demonstration policy is very coarse, and we argument the data with additional linear interpolation between consecutive states. We also use only a subset of the states as inputs to the value function and the dynamics model,

---

[10]Available at `https://github.com/openai/gym/tree/master/gym/envs/robotics`.

which apparently helps improve the training and generalization of them. Implementation details can be found in Section D.2.

Our main results are reported in Table 1 [11] for the setting with no environment interaction and Figure 3 for the setting with environment interactions. Table 1 shows that the Reach environment is too simple so that we do not need to run the RL algorithm. On the harder environments Pick-And-Place and Push, our algorithm VINS outperforms BC. We believe this is because our conservatively-extrapolated value function helps correct the mistakes in the policy. Here we use 2k trials to estimate the success rate (so that the errors in the estimation is negligible), and we run the algorithms with 10 different seeds. The error bars are for 1 standard error.

|          | VINS (ours)          | BC                  |
| -------- | -------------------- | ------------------- |
| Reach 10 | $\mathbf{99.3 \pm 0.1\%}$ | $98.6 \pm 0.1\%$ |
| Pick 100 | $\mathbf{75.7 \pm 1.0\%}$ | $66.8 \pm 1.1\%$ |
| Pick 200 | $\mathbf{84.0 \pm 0.5\%}$ | $82.0 \pm 0.8\%$ |
| Push 100 | $\mathbf{44.0 \pm 1.5\%}$ | $37.3 \pm 1.1\%$ |
| Push 200 | $\mathbf{55.2 \pm 0.7\%}$ | $51.3 \pm 0.6\%$ |

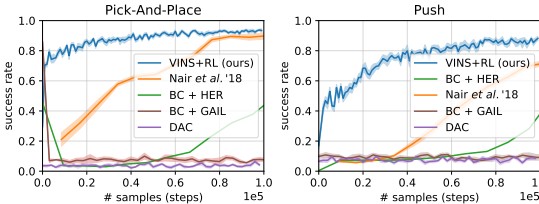

Table 1: The success rates of achieving the goals for VINS and BC in the setting without any environment interactions. A random policy has about 5% success rate at Pick and Push.

Figure 3: The learning curves of VINS+RL (Algorithm 3) vs the prior state-of-the-art Nair *et al.*'18 on Pick-And-Place and Push. Shaded areas indicates one standard error estimated from 10 random seeds.[12]

Figure 3 shows that VINS initialized RL algorithm outperforms prior state-of-the-art in sample efficiency. We believe the main reason is that due to the initialization of value and model, we pay less samples for warming up the value function. We note that our initial success rate in RL is slightly lower than the final result of VINS in Table 1. This is because in RL we implemented a slightly worse variant of the policy induced by VINS: in the policy of Algorithm 2, we use option 2 to search the action uniformly. This suffices because the additional interactions quickly allows us to learn a good model and the BC constraint is no longer needed.

**Ablation studies.** Towards understanding the effect of each component of VINS, we perform three ablative experiments to show the importance of negative sampling, searching in the neighborhood of Behavioral Cloned actions (option 1 in line 10 or Algorithm 2), and a good dynamics model. The results are shown in Table 2. We study three settings: (1) VINS without negative sampling (VINS w/o NS), where the loss $\mathcal{L}_{ns}$ is removed; (2) VINS without BC (VINS w/o BC), where option 2 in line 10 or Algorithm 2 is used; (3) VINS with oracle model without BC (VINS w/ oracle w/o BC), where we use the true dynamics model to replace line 12 of Algorithm 2. Note that the last setting is only synthetic for ablation study because in the real-world we don't have access to the true dynamics model. Please see the caption of Table 2 for more interpretations. We use the same set of hyperparameters for the same environment, which may not be optimal: for example, with more expert trajectories, the negative sampling loss $\mathcal{L}_{ns}$, which can be seen as a regularziation, should be assigned a smaller coefficient $\mu$.

## 7 CONCLUSION

We devise a new algorithm, VINS, that can learn self-correctable by learning value function and dynamical model from demonstrations. The key idea is a theoretical formulation of conservatively-extrapolated value functions that provably leads to self-correction. The empirical results show a promising performance of VINS and an algorithm that initializes RL with VINS. It's a fascinating direction to study other algorithms that may learn conservatively-extrapolated value functions in

---

[11]The standard error in the paper means the standard error of average success rate over 10 (100 for Reach 10) different random seeds by the same algorithm, that is, the standard deviation of 10 numbers over $\sqrt{10}$ (or 10, respectively).

[12]The curve for Nair et al.'s only starts after a few thousands steps because the code we use `https://github.com/jangirrishabh/Overcoming-exploration-from-demos` only evaluates after the first epoch.

|  | Pick 100 | Pick 200 | Push 100 | Push 200 |
|---|---|---|---|---|
| BC | $66.8 \pm 1.1\%$ | $82.0 \pm 0.8\%$ | $37.3 \pm 1.1\%$ | $51.3 \pm 0.6\%$ |
| VINS | $75.7 \pm 1.0\%$ | $84.0 \pm 0.5\%$ | $44.0 \pm 0.8\%$ | $55.2 \pm 0.7\%$ |
| VINS w/o BC | $28.5 \pm 1.1\%$ | $43.6 \pm 1.2\%$ | $14.3 \pm 0.5\%$ | $24.9 \pm 1.3\%$ |
| VINS w/ oracle w/o BC | $51.4 \pm 1.4\%$ | $62.3 \pm 1.1\%$ | $40.7 \pm 1.4\%$ | $42.9 \pm 1.3\%$ |
| VINS w/ oracle | $76.3 \pm 1.4\%$ | $87.0 \pm 0.7\%$ | $48.7 \pm 1.2\%$ | $63.8 \pm 1.3\%$ |
| VINS w/o NS | $48.5 \pm 2.1\%$ | $71.6 \pm 0.9\%$ | $29.3 \pm 1.2\%$ | $38.7 \pm 1.5\%$ |

Table 2: Ablation study of components of VINS in the setting without environment interactions. We reported the average performance of 10 runs (with different random seeds) and the empirical standard error of the estimator of the average performance. The success rate of VINS w/o NS is consistently worse than VINS, which suggests that NS is crucial for tackling the false extrapolation. From comparisons between VINS w/o BC and VINS w/ oracle w/o BC, and between VINS and VINS w/ oracle, we observe that if the learning of the dynamics can be improved (potentially by e.g., by collecting data with random actions), then VINS or VINS w/o BC can be improved significantly. We also suspect that the reason why we need to search over the neighborhood of BC actions is that the dynamics is not accurate at state-action pairs far away from the demonstration set (because the dynamics is only learned on the demonstration set.)

other real-world applications beyond the proof-of-concepts experiments in this paper. For example, the negative sampling by Gaussian perturbation technique in this paper may not make sense for high-dimensional pixel observation. The negative sampling can perhaps be done in the representation space (which might be learned by unsupervised learning algorithms) so that we can capture the geometry of the state space better.

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

## A  DEGENERACY OF LEARNING $Q$ FUNCTIONS FROM DEMONSTRATIONS AND OUR SOLUTIONS

Fitting $Q^{\pi_e}$ from only demonstration is problematic: there exists a function $Q(s, a)$ that does not depend on $a$ at all, which can still match $Q^{\pi_e}$ on all possible demonstration data. Consider $Q(s, a) \triangleq Q^{\pi_e}(s, \pi_e(s))$. We can verify that for any $(s, a)$ pair in the demonstrations satisfying $a = \pi_e(s)$, it holds that $Q(s, a) = Q^{\pi_e}(s, a)$. However, $Q(s, a)$ cannot be accurate for other choices of $a$'s because by its definition, it does not use any information from the action $a$.

## A.1 COPING WITH THE DEGENERACY WITH LEARNED DYNAMICAL MODEL

Cautious reader may realize that the degeneracy problem about learning $Q$ function from demonstrations with deterministic policies may also occur with learning the model $M$. However, we will show that when the problem has the particular structure of reaching a random but given goal state $g$, learning $Q$ still suffers the degeneracy but learning the dynamics does not.

The typical way to deal with a random goal state $g$ is to consider goal-conditioned value function $V(s, g)$, policy $\pi(s, g)$, and $Q$ function $Q(s, a, g)$.[13] However, the dynamical model does not have to condition on the goal. Learning $Q$ function still suffers from the degeneracy problem because $Q(s, a, g) \triangleq Q^{\pi_e}(s, \pi_e(s, g), g)$ matches $Q^{\pi_e}$ on the demonstrations but does not use the information from $a$ at all. However, learning $M$ does not suffer from such degeneracy because given a single state $s$ in the demonstration, there is still a variety of action $a$ that can be applied to state $s$ because there are multiple possible goals $g$. (In other words, we cannot construct a pathological $M(s, a) = M^\star(s, \pi_e(s))$ because the policy also takes in $g$ as an input). As a result, parameterizing $Q$ by $Q(s, a, g) = V(M(s, a), g)$ do not suffer from the degeneracy either.

## B  MISSING PROOFS IN SECTION 4

*Proof of Lemma 4.5.* Let $\bar{s} = \Pi_{\mathcal{U}}s$ and $\bar{a} = \pi_e(\bar{s})$. Because $s$ is $\varepsilon$-close to the set $\mathcal{U}$, we have $\|s - \bar{s}\| \leq \varepsilon$. By the $(\gamma, L_c, \varepsilon)$-locally-correctable assumption of the dynamics, we have that there exists an action $a_{\mathrm{cx}}$ such that a) $\|a_{\mathrm{cx}} - \bar{a}\| \leq L_c\varepsilon$ and b) $s'_{\mathrm{cx}} \triangleq M^\star(s, a_{\mathrm{cx}})$ is $\gamma\varepsilon$-close to the set $\mathcal{U}$. Next we show that $a_{\mathrm{cx}}$ belongs to the constraint set $\{a : \|a - \pi_{\mathrm{BC}}(s)\| \leq \zeta\}$ in equation (4.3). Note that $\|a_{\mathrm{cx}} - \pi_{\mathrm{BC}}(s)\| \leq \|a_{\mathrm{cx}} - \bar{a}\| + \|\bar{a} - \pi_{\mathrm{BC}}(\bar{s})\| + \|\pi_{\mathrm{BC}}(\bar{s}) - \pi_{\mathrm{BC}}(s)\| \leq L_c\varepsilon + \delta_\pi + L_\pi\varepsilon$ because of triangle inequality, the closeness of $a_{\mathrm{cx}}$ and $\bar{a}$, the assumption that $\pi_{\mathrm{BC}}$ has $\delta$ error in the demonstration state set $\mathcal{U}$, and the Lipschitzness of $\pi_{\mathrm{BC}}$. Since $\zeta$ is chosen to be bigger than $L_c\varepsilon + \delta_\pi + L_\pi\varepsilon$, we conclude that $a_{\mathrm{cx}}$ belongs to the constraint set of the optimization in equation (4.3).

This suggests that the maximum value of the optimization (4.3) is bigger than the corresponding value of $a_{\mathrm{cx}}$:

$$V(M(s, a)) \geq V(M(s, a_{\mathrm{cx}})) \tag{B.1}$$

Note that $a$ belongs to the constraint set by definition and therefore $\|a - a_{\mathrm{cx}}\| \leq 2\zeta$. By Lipschitzness of the dynamical model, and the value function $V^{\pi_e}$, we have that $\|M^\star(s, a) - M^\star(s, a_{\mathrm{cx}})\| \leq L_M\|a - a_{\mathrm{cx}}\| \leq 2L_M\zeta$. Let $s' = M^\star(s, a)$ and $s'_{\mathrm{cx}} = M^\star(s, a_{\mathrm{cx}})$. We have $\|s' - s'_{\mathrm{cx}}\| \leq 2L_M\zeta$. By the Lipschitz projection assumption, we have that $\|\Pi_{\mathcal{U}}s' - \Pi_{\mathcal{U}}s'_{\mathrm{cx}}\| \leq L_\Pi\|s' - s'_{\mathrm{cx}}\| \leq 2L_\Pi L_M\zeta$, which in turns implies that $|V^{\pi_e}(\Pi_{\mathcal{U}}s') - V^{\pi_e}(\Pi_{\mathcal{U}}s'_{\mathrm{cx}})| \leq 2L_V L_\Pi L_M\zeta$ by Lipschitzness of $V^{\pi_e}$. It follows that

$$
\begin{aligned}
V(s') &\leq V^{\pi_e}(\Pi_{\mathcal{U}}s') - \lambda\|s' - \Pi_{\mathcal{U}}s'\| + \delta_V && \text{(by assumption (4.1))}\\
&\leq V^{\pi_e}(\Pi_{\mathcal{U}}s'_{\mathrm{cx}}) + |V^{\pi_e}(\Pi_{\mathcal{U}}s'_{\mathrm{cx}}) - V^{\pi_e}(\Pi_{\mathcal{U}}s')| - \lambda\|s' - \Pi_{\mathcal{U}}s'\| + \delta_V \\
&&& \text{(by triangle inequality)}\\
&\leq V^{\pi_e}(\Pi_{\mathcal{U}}s'_{\mathrm{cx}}) + 2L_V L_\Pi L_M\zeta - \lambda\|s' - \Pi_{\mathcal{U}}s'\| + \delta_V && \text{(by equations in paragraph above)}\\
&\leq V(s'_{\mathrm{cx}}) + \lambda\|s'_{\mathrm{cx}} - \Pi s'_{\mathrm{cx}}\| + 2L_V L_\Pi L_M\zeta - \lambda\|s' - \Pi_{\mathcal{U}}s'\| + 2\delta_V && \text{(by assumption (4.2))}
\end{aligned}
$$

Note that by the Lipschitzness of the value function and the assumption on the error of the dynamical model,

$$
\begin{aligned}
|V(s') - V(M(s, a))| &= |V(M^\star(s, a)) - V(M(s, a))| \\
&\leq L_v\|M^\star(s, a) - M(s, a)\| \leq L_V\delta_M
\end{aligned} \tag{B.2}
$$

Simlarly

$$
\begin{aligned}
|V(s'_{\mathrm{cx}}) - V(M(s, a_{\mathrm{cx}}))| &= |V(M^\star(s, a_{\mathrm{cx}})) - V(M(s, a_{\mathrm{cx}}))| \\
&\leq L_v\|MT^\star(s, a_{\mathrm{cx}}) - M(s, a_{\mathrm{cx}})\| \leq L_V\delta_M
\end{aligned} \tag{B.3}
$$

---

[13]This is equivalent to viewing the random goal $g$ as part of an extended state $\check{s} = (s, g)$. Here the second part of the extended state is randomly chosen during sampling the initial state, but never changed by the dynamics. Thus all of our previous work does apply to this situation via this reduction.

Combining the three equations above, we obtain that

$$\lambda \|s'_{\text{cx}} - \Pi s'_{\text{cx}}\| + 2L_V L_\Pi L_M \zeta - \lambda \|s' - \Pi_{\mathcal{U}} s'\| + 2\delta_V$$
$$\geq V(s') - V(s'_{\text{cx}})$$
$$\geq V(M(s, a)) - V(M(s, a_{\text{cx}})) - 2L_V \delta_M \qquad \text{(by equation (B.2) and (B.3))}$$
$$\geq -2L_V \delta_M \qquad \text{(by equation (B.1))}$$

Let $\kappa = 2L_V L_\Pi L_M \zeta + 2\delta_V + 2L_V \delta_M$ and use the assumption that $s'_{\text{cx}}$ is $\gamma\varepsilon$-close to the set $\mathcal{U}$ (which implies that $\|s'_{\text{cx}} - \Pi s'_{\text{cx}}\| \leq \gamma\varepsilon$), we obtain that

$$\lambda \|s' - \Pi_{\mathcal{U}} s'\| \leq \lambda \|s'_{\text{cx}} - \Pi s'_{\text{cx}}\| + \kappa \leq \lambda\gamma\varepsilon + \kappa \qquad (B.4)$$

Note that $\lambda \geq \frac{\kappa}{(1-\gamma)\varepsilon}$, we have that $\|s' - \Pi_{\mathcal{U}} s'\| \leq \varepsilon$.

$\square$

*Proof of Theorem 4.4.* To prove bullet 1, we apply Lemma 4.5 inductively for $T$ steps. To prove bullet 2, we will prove that as long as $s_i$ is $\varepsilon$-close to $\mathcal{U}$, then we can improve the value function by at least $\rho - ?$ in one step. Consider $\bar{s}_i = \Pi_{\mathcal{U}}(s_i)$. We triangle inequality, we have that $\|\pi(s_i) - \pi_{\text{e}}(\bar{s}_i)\| \leq \|\pi(s_i) - \pi_{\text{BC}}(s_i)\| + \|\pi_{\text{BC}}(s_i) - \pi_{\text{BC}}(\bar{s}_i)\| + \|\pi_{\text{BC}}(\bar{s}_i) - \pi_{\text{e}}(\bar{s}_i)\|$. These three terms can be bounded respectively by $\zeta$, $L_\pi \|s_i - \bar{s}_i\| \leq L_\pi \varepsilon$, and $\delta_\pi$, using the definition of $\pi$, the Lipschitzness of $\pi_{\text{BC}}$, and the error assumption of $\pi_{\text{BC}}$ on the demonstration state set $\mathcal{U}$, respectively. It follows that $\|M^\star(s_i, \pi(s_i)) - M^\star(\bar{s}_i, \pi_{\text{e}}(\bar{s}_i))\| \leq L_{M,s} \|s_i - \bar{s}_i\| + L_{M,a} \|\pi(s_i) - \pi_{\text{e}}(\bar{s}_i)\| \leq L_{M,s}\varepsilon + L_{M,a}(\zeta + L_\pi \varepsilon + \delta_\pi)$. It follows by the Lipschitzness of the projection that

$$\|\bar{s}_{i+1} - \Pi_{\mathcal{U}} M^\star(\bar{s}_i, \pi_{\text{e}}(\bar{s}_i))\| = \|\Pi_{\mathcal{U}} M^\star(s_i, \pi(s_i)) - \Pi_{\mathcal{U}} M^\star(\bar{s}_i, \pi_{\text{e}}(\bar{s}_i))\| \qquad (B.5)$$
$$= |\Pi_{\mathcal{U}} M^\star(s_i, \pi(s_i)) - M^\star(\bar{s}_i, \pi_{\text{e}}(\bar{s}_i)) \qquad (B.6)$$
$$\leq L_\Pi(L_{M,s}\varepsilon + L_{M,a}(\zeta + L_\pi \varepsilon + \delta_\pi)) \qquad (B.7)$$

This implies that

$$|V^{\pi_{\text{e}}}(\bar{s}_{i+1}) - V^{\pi_{\text{e}}}(\Pi_{\mathcal{U}} M^\star(\bar{s}_i, \pi_{\text{e}}(\bar{s}_i)))| \leq L_V L_\Pi(L_{M,s}\varepsilon + L_{M,a}(\zeta + L_\pi \varepsilon + \delta_\pi)) \qquad (B.8)$$

Note that we assumed that $V(M^\star(\bar{s}_i, \pi_{\text{e}}(\bar{s}_i))) \geq V(\bar{s}_i) + \rho$ or $M^\star(\bar{s}_i, \pi_{\text{e}}(\bar{s}_i))$ reaches the goal. If the former, it follows that $V^{\pi_{\text{e}}}(\bar{s}_{i+1}) \geq V^{\pi_{\text{e}}}(\bar{s}_i) + \rho - L_V L_\Pi(L_{M,s}\varepsilon + L_{M,a}(\zeta + L_\pi \varepsilon + \delta_\pi)) \geq V^{\pi_{\text{e}}}(\bar{s}_i) + \rho/2$. Otherwise, or $s_{i+1}$ is $\varepsilon$-close to $\bar{s}_{i+1}$ whose value is at most $-L_V L_\Pi(L_{M,s}\varepsilon + L_{M,a}(\zeta + L_\pi \varepsilon + \delta_\pi)) = -O(\varepsilon + \delta_\pi)$

$\square$

# C    ALGORITHMS VINS+RL

A pseudo-code our algorithm VINS+RL can be found in Algorithm 3

---

**Algorithm 3** Value Iteration with Environment Interactions Initialized by VINS (VINS+RL)

---

**Require:** Initialize parameters $\phi, \theta$ from the result of VINS (Algorithm 2)
1: $\mathcal{R} \leftarrow$ demonstration trajectories;
2: **for** stage $t = 1, \dots$ **do**
3:     collect $n_1$ samples using the induced policy $\pi$ in Algorithm 2 (with Option 2 in Line 10) and add them to $\mathcal{R}$
4:     **for** $i = 1, \dots, n_{\text{inner}}$ **do**
5:         sample mini-batch $\mathcal{B}$ of $N$ transitions $(s, a, r, s')$ from $\mathcal{R}$
6:         update $\phi$ to minimize $\mathcal{L}_{td}(\phi; \mathcal{B})$
7:         update target value network: $\bar{\phi} \leftarrow \bar{\phi} + \tau(\phi - \bar{\phi})$
8:         update $\theta$ to minimize loss $\mathcal{L}_{\text{model}}(\theta; \mathcal{B})$

---

# D IMPLEMENTATION DETAILS

## D.1 SETUP

In the three environments, a 7-DoF robotics arm is manipulated for different goals. In Reach, the task is to reach a randomly sampled target location; In Pick-And-Place, the goal is to grasp a box in a table and reach a target location, and in Push, the goal is to push a box to a target location.

The reward function is 0 if the goal is reached; otherwise -1. Intuitively, an optimal agent should complete the task in a shortest possible path. The environment will stop once the agent achieves the goal or max step number have been reached. Reaching max step will be regarded as failure.

For more details, we refer the readers to (Plappert et al., 2018).

## D.2 HYPERPARAMETERS

**Behavioral Cloning** We use a feed-forward neural network with 3 hidden layers, each containinig 256 hidden units, and ReLU activation functions. We train the network until the test success rate plateaus.

**Nair et al.'18 (Nair et al., 2018)** We use the implementation from `https://github.com/ jangirrishabh/Overcoming-exploration-from-demos`. We don't change the default hyperparameters, except that we're using 17 CPUs.

**GAIL (Ho & Ermon, 2016)** We use the implementation from OpenAI Baselines (Dhariwal et al., 2017). We don't change the default hyperparameters.

**HER (Andrychowicz et al., 2017)** We also use the code from OpenAI Baselines and keep the default hyperparameters.

**Discriminator-Actor-Critic (Kostrikov et al., 2018)** We use the implementation from the official implementation `https://github.com/google-research/google-research/ tree/master/dac`.

**VINS**

- Architecture: We use feed-forward neural networks as function approximators for values and dynamical models. For the $V_\phi$, the network has one hidden layer which has 256 hidden units and a layer normalization layer (Lei Ba et al., 2016). The dynamics model is a feed-forward neural network with two hidden layers and ReLU activation function. Each hidden layer has 500 units. The model uses the reduced states and actions to predict the next reduced states.

- Value augmentation: We augment the dataset by a linear interpolation between two consecutive states, i.e., for a transition $(s, a, r, s')$ in the demonstration, it's augmented to $(s + \lambda(s' - s), a, \lambda r, s')$ for a random real $\lambda \sim \text{Uniform}[0, 1]$. To minimize the losses, we use the Adam optimizer Kingma & Ba (2014) with learning rate $3 \times 10^{-4}$. We remove some less relevant coordinates from the state space to make the dimension smaller. (But we maintain the full state space for BC. BC will perform worse with reduce state space.) Specifically, the states of our algorithm for different environment are a) Reach: the position of the arm, b) Pick: the position of the block, the gripper, and the arm, and c) Push: the position of the block and the arm.

- States representation and perturbation: Both our model $M$ and value function $V$ are trained on the space of reduced states. The perturb function is also designed separately for each task. For Reach and Push, we perturb the arm only; For Pick-And-Place, we perturb the arm or the gripper. In the implementation, we perturb the state $s$ by adding Gaussian noise from a distribution $\mathcal{N}(0, \rho\Sigma)$, where $\Sigma$ is a diagonal matrix that contains the variances of each coordinate of the states from the demonstration trajectories. Here $\rho > 0$ is a hyper-parameter to tune.

|  | Pick 100 | Pick 200 | Push 100 | Push 200 |
|---|---|---|---|---|
| BC | $66.8 \pm 1.1\%$ | $82.0 \pm 0.8\%$ | $37.3 \pm 1.1\%$ | $51.3 \pm 0.6\%$ |
| VINS | $75.7 \pm 1.0\%$ | $84.0 \pm 0.5\%$ | $44.0 \pm 0.8\%$ | $55.2 \pm 0.7\%$ |
| BC w/ data aug | $58.8 \pm 1.3\%$ | $77.4 \pm 0.9\%$ | $30.1 \pm 0.8\%$ | $41.4 \pm 1.6\%$ |
| VINS w/ oracle w/o NS | $47.2 \pm 1.9\%$ | $74.3 \pm 1.0\%$ | $30.7 \pm 0.8\%$ | $41.3 \pm 0.8\%$ |
| VINS w/ oracle | $76.3 \pm 1.4\%$ | $87.0 \pm 0.7\%$ | $48.7 \pm 1.2\%$ | $63.8 \pm 1.3\%$ |
| VINS w/o NS | $48.5 \pm 2.1\%$ | $71.6 \pm 0.9\%$ | $29.3 \pm 1.2\%$ | $38.7 \pm 1.5\%$ |

Table 3: Ablation study of components of VINS in the setting without environment interactions. We reported the average performance of 10 runs (with different random seeds) and the empirical standard error of the estimator of the average performance. The success rate of BC w/ data augmentation is consistently worse than BC w/o data augmentation. )

## E  ADDITIONAL EXPERIMENTS

We have some additional experiments to evaluate VINS, which are (a) BC with data augmentation (b) VINS with oracle without negative sampling, which are explained below. The results are summarized in Table 3.

**BC w/ data augmentation.**   As VINS augments the data to train the value function, it might be interesting to see the performance of BC w/ data augmentation to make the comparison more fair. The way we augment the data is similar to what we have done for the value function: Given two consecutive state-action pairs $(s, a)$ and $(s', a')$, where $s' = M(s, a)$, we sample $t \sim \text{Uniform}[0, 1]$ and construct a new pair $(ts + (1 - t)s', ta + (1 - t)a')$. The new pair is used for behavior cloning. However, we found out that this kind of data augmentation hurts. We hypothesize that the augmented data provide incorrect information, as the actions between two consecutive states might be very non-linear, while value is more linear. We suspect that the biases/errors introduced by the data augmentation is particularly harmful for BC, because it lacks a mechanism of self-correction that VINS has.

**VINS w/ oracle w/o negative sampling.**   Another way to isolate the effect of negative sampling is comparing VINS w/ oracle w/o NS to VINS w/ oracle. The results are shown at Table 3. We can see that VINS w/ oracle achieves a significantly higher success rate than VINS w/ oracle w/o negative sampling, which shows that negative sampling is essential to the success of VINS.

**VINS with fewer demonstrations.**   To evaluate whether VINS works with few demonstrations, we also conduct our experiments wih only 40 expert trajectories. The result is shown in Table 4. The performance of BC is worse than that with 100 expert trajectories, VINS can still correct some actions via learned value function and model and achieves a higher success rate.

**VINS with stochatic dynamics model.**   As this paper mainly studies deterministic dynamics model, we'd like to note that our theory can be generalized to stochastic dynamics model easily. We also conduct proof-of-concept experiments to show the performance of VINS in this case. The stochastic environment we used in the experiments is based on Push, and the modification is that the dynamics model adds noises to the received actions, i.e., $s' \sim M(s, a + \zeta)$ where $\zeta \sim \mathcal{N}(0, \sigma^2 I)$, $M$ here is the original deterministic dynamics model. One intuitive interpretation is that the actuator on the robot might not be good enough to produce the exact forces. For simplicity we don't train a stochastic dynamics model but learn a deterministic one to approximate it. The result is summarized in Table 4, in which we can find even with a determinisic dynamics model, VINS performs better than BC.

|  | VINS (ours) | BC |
|---|---|---|
| Pick 40 | $40.0 \pm 0.9\%$ | $36.3 \pm 1.7\%$ |
| Push 40 | $26.5 \pm 0.6\%$ | $23.5 \pm 0.7\%$ |
| StochasticPush 100 | $38.4 \pm 0.9\%$ | $29.6 \pm 1.4\%$ |
| StochasticPush 200 | $51.5 \pm 0.7\%$ | $42.5 \pm 0.9\%$ |

Table 4: The success rates of achieving the goals for VINS and BC in the setting without any environment interactions. We reported the average performance of 10 runs (with different seeds) and the empirical standard error of the esitmator of the average performance. VINS outperforms BC in all tested environments.

