# OpenReview forum: "Learning Self-Correctable Policies and Value Functions from Demonstrations with Negative Sampling"
_ICLR.cc/2020/Conference — Accept (Poster)_

### Official Review · AnonReviewer1 · 2019-10-21
**Official Blind Review #1**

**Rating:** 6

**Review:**

This work tried to address the covariate shift problem in imitation learning, which is due to the mismatch between training and test state distribution and may cause compounding errors.

The authors proposed the algorithm called value iteration with negative sampling (VINS) of which the main ideas can be summarized as follows. First, value iteration is used on expert trajectories with negative sampling. Specifically, the states that are randomly perturbed from expert’s states were used to enforce (4.1) and (4.2) in the submission (called *conservative extrapolation* requirements in the submission). By doing so, the state values outside the support of expert state visitation distribution become less than those inside the support. In the meantime, temporal-difference (TD) error was minimized to satisfy Bellman consistency among state values. The second main idea of this work is to use *self-correctable policy*, where the approximate dynamics and behavioral-cloning (BC) policy were used to select the action which is expected to give higher value at the successor states.

To consolidate their ideas, the authors proved Theorem 4.4 showing that state visitation distribution of resultant policy is approximately close to that of an expert under a few assumptions. Empirically, they considered two experiments. In the first experiment, assuming that the environment simulation is not allowed, the performance of VINS was compared with that of BC over 5 tasks, and VINS achieved higher success rates. In the second experiment, assuming the simulation is allowed, VINS was compared with existing methods such as HER+BC, GAIL and Nair et al ‘18 and shown to be much more sample-efficient compared to the selected baselines.

Although the theoretical and empirical contributions of this work are clear to me when the environment simulation is not allowed (as shown in the first experiment in Table 1), I think the second experiment, which allows the environment simulation (as shown in Figure 3), is misleading, and this is why I vote weak reject for this work. For instance, we can simply think about GAIL with BC initialization, but it seems to me that GAIL with random initialization was used in the second experiment (since authors mentioned GAIL in OpenAI baselines was used without hyperparameter tuning (https://github.com/openai/baselines/blob/master/baselines/gail/run_mujoco.py#L53)). In addition to it, there have been some recent works on the sample efficiency of imitation learning with environment simulation which are not included as baselines in this work:

[1] GMMIL (Kim and Park, 2018, “Imitation Learning via Kernel Mean Embedding”) - cost learning with maximum mean discrepancy minimization leads to sample-efficient training

[2] BGAIL (Jeon et al, 2019, “Bayesian Approach to Generative Adversarial Imitation Learning”) - Bayesian cost is helpful for sample-efficient imitation learning

[3] DAC (Kostrikov et al, 2019, “Discriminator-Actor-Critic: Addressing Sample Inefficiency and Reward Bias in Adversarial Imitation Learning”) - Solving reward bias in imitation learning with simulation and using off-policy RL algorithm to enhance the sample efficiency

[4] Sasaki et al, 2019, “Sample Efficient Imitation Learning for Continuous Control” - Bypassing cost learning and introducing off-policy RL to enhance the sample efficiency.

Especially, off-policy imitation learning methods [3], [4] are shown to be extremely sample-efficient compared to GAIL. I think the authors should have compared the performance of VINS + RL with those baselines in the second experiment if they tried to emphasize the sample efficiency of VINS + RL. Otherwise, they should have focused more on the initialization effect of VINS and BC. For example, one can consider the convergence speed of GAIL to the expert performance when policies were initialized with either VINS or BC.

**Experience Assessment:**

I have published one or two papers in this area.

**Review Assessment: Checking Correctness Of Derivations And Theory:**

I assessed the sensibility of the derivations and theory.

**Review Assessment: Checking Correctness Of Experiments:**

I carefully checked the experiments.

**Review Assessment: Thoroughness In Paper Reading:**

I read the paper at least twice and used my best judgement in assessing the paper.

---

> ### Author Response · Authors · 2019-11-15
> **Response to R1**
>
> We thank the reviewer for the helpful comments. The review noted that VINS “achieved higher success rates” and is “shown to be much more sample-efficient compared to the selected baselines”. We have updated with the references that the reviewer kindly suggested, and added additional experiments in the revision.  We’ll address the comments below.
>
> ---  “Although the theoretical and empirical contributions of this work are clear to me when the environment simulation is not allowed (as shown in the first experiment in Table 1), I think the second experiment, which allows the environment simulation (as shown in Figure 3), is misleading, and this is why I vote weak reject for this work. For instance, we can simply think about GAIL with BC initialization, but it seems to me that GAIL with random initialization was used in the second experiment (since authors mentioned GAIL in OpenAI baselines was used without hyperparameter tuning (https://github.com/openai/baselines/blob/master/baselines/gail/run_mujoco.py#L53)).
> “
>
> In the work of Sasaki et al, 2019 [4], section 5.3, the authors have empirically showed, somewhat surprisingly, that BC-initialized GAIL does work not better than BC itself. We also run the experiment on our tasks and find out that the performance of BC initialized GAIL is also not better than vanilla GAIL. Please see Figure 3 of the revision for the curve of BC+GAIL.
>
> In general, we do think the baselines that we compare are a set of strong baselines for the manipulation tasks that we are working on. We also note that because we are working with the setting where a sparse reward function is available when environment interaction is allowed, there are a broader set of RL algorithms to use with environment interactions beyond GAIL-related approaches.  We focus on the sparse reward setting because we believe that in many real-world robotic applications a sparse reward is available, and the main goal of imitation learning in this case is to improve the sample efficiency.  In this setting,  GAIL is apparently not very competitive (partly because it does not use the sparse reward). This is also a reason why we are not centering  our work around comparing with GAIL (and its related work), because we are using more information than GAIL uses (and thus achieve better performance). Instead, we focus more on comparing with the strong baselines reported in Nair et al.
>
> ---  “I think the authors should have compared the performance of VINS + RL with those baselines in the second experiment if they tried to emphasize the sample efficiency of VINS + RL. Otherwise, they should have focused more on the initialization effect of VINS and BC. For example, one can consider the convergence speed of GAIL to the expert performance when policies were initialized with either VINS or BC. In addition to it, there have been some recent works on the sample efficiency of imitation learning with environment simulation which are not included as baselines in this work. ”
>
> Thanks for the suggestion of the references and baselines. We have cited these papers. Following the suggestion, we run the official code of  the DAC algorithm in [3] on our environments. The result is formally reported in Figure 3 of the revision. Unfortunately, DAC couldn’t get any non-trivial reward within 100K samples.
>
> We would like to note that VINS cannot and perhaps should not be used to initialize GAIL because the strength of VINS is to provide a good initial value function. If we only use the policy learned from VINS to initialize GAIL or other RL algorithms, then very likely it won’t be better than using BC to initialize them. The strength of VINS is that it  provides a good value function and dynamics so that a value-based RL algorithm can be initialized with a good initialization of value function, Q-function, and policy. Please see Appendix C for the precise RL algorithm we use, and how we make it compatible with VINS. We cannot initialize the policy of a generic RL algorithm to see the benefit of VINS. We consider this as the main contribution of VINS --- it allows us to fully initialize value-based RL algorithms.

---

### Official Review · AnonReviewer2 · 2019-10-23
**Official Blind Review #2**

**Rating:** 6

**Review:**

This work presents the value iteration with negative sampling (VINS) algorithm, a method for accelerating reinforcement learning using expert demonstrations.  In addition to learning an expert policy through behavioral cloning, VINS learns an initial value function which is biased to assign smaller expected values to states not encountered during demonstrations.  This is done by augmenting the demonstration data with states that have been randomly perturbed, and penalizing the value targets for these states by a factor proportional to their Euclidean distance to the original state.  In addition to the policy and value function, VINS also learns a one-step dynamics model used to select actions against the learned value function.  As the value function learned in VINS is only defined with respect to the current state, action values are estimated by sampling future states using the learned model, and computing the value of these sampled states.

Empirical results on a set of robotic control tasks demonstrate that VINS requires significantly less interaction with the environment to learn a good policy than existing, state of the art approaches given the same set of demonstrations.  While the paper presents a novel and highly effective approach, there are some apparent limitations to the algorithms which should be highlighted, and there is room for improvement in the empirical evaluations.

It is unclear that VINS would generalize well beyond robotic control domains.  For one, its theoretical guarantees depend on the local reversibility of the dynamics, that is, for small deviations from the desired state, it is possible to return to that state in a single step.  This isn't too significant a restriction, as the ability to recover quickly from small mistakes would seem to be a necessary for any method to be able to provide similar guarantees about its behavior.  The bigger issue is the use of the Euclidean metric (or any fixed metric) in the definition of conservative extrapolation.  Basically, a state is said to be similar to the states observed during the demonstrations if it is close, under the Euclidean metric, to at least one demonstrated state.  This is a reasonable approach in robotic control tasks, where Euclidean distance is a good measure of how similar two configurations are to one another, but it would seem to be unsuitable for domains where the observation space consists of images or other high-dimensional representations.  In those cases, a useful notion of similarity would likely have to be learned from the data.  In such domains, one might imagine that the conservative value function would simply learn to distinguish between real observations, and those that have been perturbed by random noise, which would never be observed in the actual task.

While experimental results demonstrate a very significant advantage for VINS both in terms of sample complexity and final performance, results are presented for only two tasks, 'pick-and-place' and 'push', while VINS outperforms the alternatives on these tasks, it is worth noting that its initial performance (without additional environment interact) is not dramatically superior to pure behavioral cloning.  It would be helpful to see how well VINS compares against the alternatives for a much smaller number of demonstrations, say 5-20, a regime where we would expect initial performance to be poor.

**Experience Assessment:**

I have read many papers in this area.

**Review Assessment: Checking Correctness Of Derivations And Theory:**

I assessed the sensibility of the derivations and theory.

**Review Assessment: Checking Correctness Of Experiments:**

I assessed the sensibility of the experiments.

**Review Assessment: Thoroughness In Paper Reading:**

I read the paper at least twice and used my best judgement in assessing the paper.

---

> ### Author Response · Authors · 2019-11-15
> **Response to R2**
>
> We thank the anonymous reviewer 2 for the helpful comments. The review noted that our method is “a novel and highly effective approach” and “has a very significant advantage both in terms of sample complexity and final performance”. The reviewer also has many constructive questions and comments which we will address below.
>
>
> > “For one, its theoretical guarantees depend on the local reversibility of the dynamics, that is, for small deviations from the desired state, it is possible to return to that state in a single step.  This isn't too significant a restriction, as the ability to recover quickly from small mistakes would seem to be a necessary for any method to be able to provide similar guarantees about its behavior.  “
>
> We agree with the reviewer on the point that the local-correctability is key to VINS and is suitable for robotic control problems.
>
> > “It is unclear that VINS would generalize well beyond robotic control domains. … The bigger issue is the use of the Euclidean metric (or any fixed metric) in the definition of conservative extrapolation.”
> We agree with the reviewer Euclidean metric are not applicable to certain situations such as pixel space. We design the algorithm to learn self-correctable policies in the robotic control domain. Indeed, we do not expect the technique can directly work in a pixel space as it is.  However, we believe that applying perturbation to the latent representation of images would be a very  promising and interesting direction for future work. Concretely, it’s conceivable that the metric can be the Euclidean distance between the latent representations of the states. (Our theory still applies to this metric, though the metric needs to be learnt.)
> > “It would be helpful to see how well VINS compares against the alternatives for a much smaller number of demonstrations, say 5-20, a regime where we would expect initial performance to be poor.”
>
> Thanks for the suggestions. For the Reach environment, we run our algorithm on 10 demonstration trajectories. And the result is:
>
> VINS = 99.3 +/- 0.1%, BC = 98.6 +/- 0.1%.
>
> For Push and PickAndPlace environment, we found BC (and VINS) have very low success rate (below 20%) with less than 20 trajectories, and therefore we suspect the results are not very meaningful.  Thus, we conducted experiments on 40 demonstration trajectories. The results are:
>
> PickAndPlace-40: VINS = 40.0 +/- 0.9%, BC = 36.3 +/- 1.7%
> Push-40: VINS = 26.5 +/- 0.6%, BC = 23.5 +/- 0.7%
>
> (Please see Table 4 in Appendix E of the revision for a formal table.)

---

### Official Review · AnonReviewer3 · 2019-10-24
**Official Blind Review #3**

**Rating:** 6

**Review:**


This paper tackles an issue imitation learning approaches face. More specifically, policies learned in this manner can often fail when they encounter new states not seen in demonstrations. The paper proposes a method for learning value functions that are more conservative on unseen states, which encourages the learned policies to stay within the distribution of training states. Theoretical results are derived to provide some support for the approach. A practical algorithm is also presented and experiments on continuous control tasks display the effectiveness of the method, with particularly good results on imitation learning followed by reinforcement learning.

The proposed algorithm makes use of a natural intuition, that states visited by the expert probably have higher values, and the paper generally does a good job of supporting the approach through theory and experiments. Although the experiments seem sound, certain experimental details are not completely clear. The theory may also have some restrictive assumptions, limiting its significance.
Overall, I am divided about this paper. While this submission has the elements of a good paper and the presentation is great, certain concerns make me hesitant to recommend acceptance. I would be willing to increase my score if those points are addressed.

Theory:
1) My main concern is the applicability of the theorem, due to Assumption 4.2. While the intuition is that there is an action that corrects the next state towards the demonstration states, the theoretical condition is more restrictive. In particular, the following part (paraphrasing): "there exists an action a_cx that is close to a^bar and that makes a correction towards U". This condition implies that there are correcting actions near any action a^bar, which sems unrealistic in most cases. For example, in a driving task, say s^bar is a state such that moving back to U require the vehicle to move to the left. Then, consider the action a^bar of steering towards the right (with some angle). There could be no action near a^bar that makes the vehicle turn left towards U. Note that this is not necessarily a pathological situation as described in the text.

2) Also concerning assumption 4.2, I do not see why s^bar' is included in the quantifier of the statement since it is not used afterwards; after "there exists an action..." no mention of s^bar' is made.

3) The projection function may not be well-defined if there are multiple states that are closest to the one being projected.

4) It could be said explicitly that the expert policy is assumed to be deterministic. Currently, this is not said outright.

Experiments:
1) It seems like VINS relies heavily on the assumption that the environment is deterministic to learn an effective model. Was VINS tried in stochastic environments?

2) Data augmentation is used for VINS. This seems like an unfair advantage compared to the baseline competitors since sample efficiency is a key concern to reinforcement learning algorithms.  To make the comparisons fair, either it should be removed or the other algorithms should also receive additional data. How is the performance of VINS without this addition?

3) A description of how the hyperparameters were chosen and their final values would be needed for reproducibility. Also, a discussion of the importance of the hyperparameters and their sensitivity would be informative. For example, I was curious to know the value of \alpha in Algorithm 2 compared to the ranges the actions could take.

4) I am not convinced that using Q functions would necessarily fail. On p.6, the paragraph "can we learn conservatively-extrapolated Q-function" gives some reasoning why this could fail, that we may not want to penalize unseen actions. This is in opposition to the BCQ algorithm [1], which explicitly tries to avoid unseen actions and still has good performance. Trying a variant with Q(s,a) could be worthwhile.
I am not exactly sure if I understood Appendix A properly but, from my understanding, I do not think the argument made there necessarily invalidates using Q functions. It seems to apply mostly to deterministic expert policies and also Q(s,a) could still have reasonable values due to function approximation (even if the particular action 'a' is not seen in demonstrations).

5) Which RL algorithms were used for the imitation learning + RL set of experiments?

6) For table 1, are the results also averaged over different sets of demonstrations?

7) Are error bars one standard deviation or one standard error (divided by sqrt(n))?

8) For figure 3, using RL without imitation learning would serve as a good additional baseline

9) Ablation study: Trying no negative sampling with a perfect model could isolate the effect of negative sampling.

10) Ablation study: What is the no behavior cloning and perfect model experiment trying to show?

11) I think the name of the algorithm should be modified as "value iteration" refers to a specific dynamic programming algorithm for learning value functions, while the proposed algorithm does not resemble this at all.

Minor comments and typos (no impact on score):
- Using the cross-entropy method as in QTOpt [2] could be used to pick actions in a more refined manner.
- There is a large amount of blank space on p.8
- p.3 "At the test time" -> "At test time"
- p.4 "entire states space" -> "entire state space", "burned to warm up them" -> "burned to warm them up"
- p.9  "option 2 by search the action uniformly."  -> "option 2 to search the actions uniformly"

[1] "Off-Policy Deep Reinforcement Learning without Exploration" by Fujimoto et al.
[2] "QT-Opt: Scalable Deep Reinforcement Learning for Vision-Based Robotic Manipulation" by Kalashnikov et al.


**Experience Assessment:**

I have read many papers in this area.

**Review Assessment: Checking Correctness Of Derivations And Theory:**

I assessed the sensibility of the derivations and theory.

**Review Assessment: Checking Correctness Of Experiments:**

I carefully checked the experiments.

**Review Assessment: Thoroughness In Paper Reading:**

I read the paper thoroughly.

---

> ### Author Response · Authors · 2019-11-15
> **Response to R3, Part 1/3**
>
> We thank the Anonymous Reviewer 3 for the detailed and helpful comments. The review noted that “good results on imitation learning followed by reinforcement learning” and “theoretical results are derived to provide some support for the approach”. The reviewer also has many constructive questions and comments which we will address below.
>
> Regarding the theoretical part of the paper:
>
> Q1: “My main concern is the applicability of the theorem, due to Assumption 4.2. While the intuition is that there is an action that corrects the next state towards the demonstration states, the theoretical condition is more restrictive. In particular, the following part (paraphrasing): "there exists an action a_cx that is close to a^bar and that makes a correction towards U"...”
>
> We first note that we only require a local correctability,  that is, when we make a minor mistake (an action that is close but not exactly the same as the optimal action), then there is an action that can drive the state closer to the demonstration set. (Please also see the response for Q2 below as well on a more technical level.) We consider this assumption relatively mild.
>
>  We agree that the correctability assumption may not apply to every environment, but we would like to argue that it’s close to necessary for imitation learning to work. Consider an environment where it’s possible to make a minor mistake to reach a state that is not correctable. Then this environment is so non-robust that we need to follow exactly the path of the expert policy, and this seems to be fundamentally challenging for learning algorithms without any prior knowledge.
>
> Q2: “Also concerning assumption 4.2, I do not see why s^bar' is included in the quantifier of the statement since it is not used afterwards; after "there exists an action..." no mention of s^bar' is made”.
>
> The reviewer is right that $\bar{s}$ is not mentioned after “there exists an action...”, but it’s mentioned right before it in the phrase “any $\varepsilon_0$-perturbation s of $\bar{s}$” to quantify the scope of $s$. In other words, we want to assume correctability, but we don’t need correctability of every state $s$ because that might be unrealistic. Instead, we only require correctability of  a state $s$ that is close to some state $\bar{s}$ in the demonstrate set.
>
> Q3: “The projection function may not be well-defined if there are multiple states that are closest to the one being projected.”
>
> For simplicity, we can (and should) use some arbitrary tiebreaker to define the projection. We’ve clarified this in the revision.   (A slightly mathematically better but involved way would be to define the projection as the set of closest points. Then we need to measure the distance between projections by some distance between the sets, which seems unnecessarily complicated for the purpose of this paper.)
>
> Q4:  “It could be said explicitly that the expert policy is assumed to be deterministic. Currently, this is not said outright.”
>
> Thanks for pointing it out. We are indeed assuming a deterministic expert policy, and we’ve clarified this in the revision.
>
> Questions regarding the experiments:
>
> Q1: “It seems like VINS relies heavily on the assumption that the environment is deterministic to learn an effective model. Was VINS tried in stochastic environments? “
>
> To learn the value function, the implementation of VINS does not require a deterministic model (although the analysis is only for deterministic model for the moment.) With the value function, suppose we also have a good estimator for the stochastic model, we can still use a variant of 4.3 to compute the action. (Precisely, we maximize $\mathbb{E}[V(M(s,a))]$ where the randomness is from the model $M$.) However, learning a stochastic environment is often challenging, due to various issues such as calibration.
>
> We also conducted a proof-of-concept experiment on stochastic environments by modifying the Push environment into a stochastic one. (Precisely, we create a stochastic Push environment by which takes in s, and a, and returns the Push(s,a+noise) as the stochastic outcome.) For simplicity, we learn the value function the same as in VINS; but  we only learn a deterministic dynamics model  to approximate the stochastic one, because we consider learning stochastic dynamics  with good calibration is an orthogonal issue and is beyond the scope of this paper. However, even with a learned deterministic dynamics, we found that VINS outperforms BC. As shown below (or formally in Table 4 in Appendix E of the revision), on the stochastic version of Push environment, VINS outperforms BC by about 9% with with both 100 or 200 demonstration trajectories.
>
> StochasticPush-100: VINS = 38.4 +/- 0.9%, BC = 29.6 +/- 1.4%
> StochasticPush-200: VINS = 51.5 +/- 0.7%, BC = 42.5 +/- 0.9%

---

> ### Author Response · Authors · 2019-11-15
> **Response to R3, Part 2/3**
>
> Q2:  “Data augmentation is used for VINS. This seems like an unfair advantage compared to the baseline competitors since sample efficiency is a key concern to reinforcement learning algorithms.  To make the comparisons fair, either it should be removed or the other algorithms should also receive additional data. How is the performance of VINS without this addition? “
>
> Thanks for the suggestions. We think our comparison is arguably fair, because we only use data augmentation to train our value function.  BC is part of VINS, and the BC component in VINS is not trained with any data augmentation. If it were the case that BC with data augmentation can outperform the vanilla BC, then using the data-augmented version of BC will also likely improve the performance of VINS.
>
> However, unfortunately, we observed that BC cannot be improved by any data augmentation that we are aware of. Concretely, we tried the following data augmentation for BC, which is similar to that for the value function in VINS: given two consecutive state-action pairs (s, a) and (s’, a’), where s’ = M(s, a), we sample t ~ Uniform[0, 1] and construct a new data point pair (t s + (1 - t) s’, t a + (1 - t) a’) as the augmented data. After we use this data-augmentation for BC, we found out that the success rate of BC is even worse as shown below (see Table 3 in Appendix E for a formal table):
>
> PickAndPlace-100: with data aug = 58.8 +/- 1.3%, without data aug = 66.8 +/- 1.1%
> PickAndPlace-200: with data aug = 77.4 +/- 0.9%, without data aug = 82.0 +/- 0.8%
> Push-100: with data aug = 30.1 +/- 0.8%, without data aug = 37.3 +/- 1.1%
> Push-200: with data aug = 41.4 +/- 1.6%, without data aug = 51.3 +/- 0.6%
>
> We suspect that the biases/errors introduced by the data augmentation is particularly harmful for BC, because it lacks a mechanism of self-correction that VINS has.
>
>
> Q3: “A description of how the hyperparameters were chosen and their final values would be needed for reproducibility. Also, a discussion of the importance of the hyperparameters and their sensitivity would be informative. For example, I was curious to know the value of \alpha in Algorithm 2 compared to the ranges the actions could take. “
>
> For $\alpha$, we tried 0.05, 0.1 and 0.2 and found out 0.1 works best. Each entry of the action is within [-1, 1]. The results of  0.05 and 0.1 don’t differ much, while the result for 0.2 is slightly worse.  For $\lambda$, we tried 15, 20, 25, and found out 20 works best.
> We found $\lambda$ is the most sensitive hyper-parameter ---  large $\lambda$ leads to difficult optimization and small $\lambda$ leads to not enough negative sampling. We will also make the code of the paper public.
>
> Q4: “I am not convinced that using Q functions would necessarily fail. On p.6, the paragraph "can we learn conservatively-extrapolated Q-function" gives some reasoning why this could fail, that we may not want to penalize unseen actions. This is in opposition to the BCQ algorithm [1], which explicitly tries to avoid unseen actions and still has good performance… I am not exactly sure if I understood Appendix A properly but, from my understanding, I do not think the argument made there necessarily invalidates using Q functions. ”
>
> We would like to clarify that we are not saying that we shouldn’t penalize unseen actions. We are saying that those unseen actions that can correct the trajectory shouldn’t be penalized. In other words, they are two types of unseen actions. Type I is those actions that are not seen because they are completely wrong actions. These should be penalized, and that’s the key idea of BCQ and a few related approach. The second type is those actions that are only useful to correct the trajectories when the state starts to leave the demonstration set. These actions would have not been useful if all the previous actions are exactly optimal, whereas otherwise they are very useful to prevent the cascading errors. Therefore, penalizing unseen actions may be useful, but it wouldn’t encourage the self-correction behavior that this paper is aiming to learn.  We suspect that the self-correction behavior is very important for imitating perfect demonstration. (Empirically, in the early stage of the project, we have also  implemented negative sampling on Q-function and found out that it does not provide the correction effect even for the toy grid-world  environment in Figure 1.)
>
> We clarify that Appendix A is not meant to invalidate using Q-functions. The first paragraph in Appendix argues that the false extrapolation issue (without any fix) exists with Q-function as well, which is perhaps a common consensus. The section A.1 only meant to justify that our VINS idea doesn’t suffer from the same extrapolation problem, but it doesn’t invalidate the use of Q-functions or other methods that addresses the use of extrapolation issue of Q-functions. (To some extent, VINS also uses Q-functions --- the Q-function in VINS is a composition of V and M.)

---

> ### Author Response · Authors · 2019-11-15
> **Response to R3, Part 3/3**
>
> Q5: “Which RL algorithms were used for the imitation learning + RL set of experiments?”
>
> We use a simple variant of model-based value iteration algorithm, the detailed description of which is deferred to Appendix C due to page limit. We design this new algorithm so that we can use the value and the dynamics learned from VINS more easily.  (We also briefly mentioned it at the last paragraph of Section 5. )
>
> Q6: “For table 1, are the results also averaged over different sets of demonstrations? “
>
> No, we only tried one set of demonstration following the setting in  [1].
>
> Q7: “Are error bars one standard deviation or one standard error (divided by sqrt(n))?”
>
> It’s one standard error. We have made this more explicit in the revision.
>
> Q8: “For figure 3, using RL without imitation learning would serve as a good additional baseline “
>
> Thanks for the suggestion. Our understanding is that for the sparse reward tasks, the best RL algorithms are HER [2] that we used to generate the expert policy. As reported by Andrychowicz et al. [2], Figure 3, these algorithms typically require 2 million steps to achieve near optimal success rate (e.g., 95%), whereas our algorithm needs about 100K.
>
> Q9: “Ablation study: Trying no negative sampling with a perfect model could isolate the effect of negative sampling. “
>
> Thanks for the suggestions. This is indeed a very useful ablation study in addition to those in Table 2. We conducted the suggested experiment and found that indeed no negative sample (NS) with oracle mode is significantly worse than negative sample with oracle model as shown below.
> (Here PickAndPlace-100 means that we have 100 trajectories in the demonstration.)
>
> PickAndPlace-100:
> Oracle model w/o NS: 47.2 +/- 1.9%; oracle model w/  NS: 76.3 +/- 1.4%
> PickAndPlace-200:
> Oracle model w/o NS: 74.3 +/- 1.0%; oracle model w/ NS: 87.0 +/- 0.7%
> Push-100:
> Oracle model w/o NS: 30.7 +/- 0.8%; oracle model w/ NS: 48.7 +/- 1.2%
> Push-200:
> Oracle model w/o NS: 41.3 +/- 0.8%; oracle model w/NS: 63.8 +/- 1.3%
> (Please see Table 3 in Appendix E in the revision for a formal table.)
>
> Q10: “Ablation study: What is the no behavior cloning and perfect model experiment trying to show?”
>
> By comparing VINS w/ perfect  w/o BC and VINS w/o BC, we can see how a better model improves the performance (even if BC is not used.) In reality, it’s conceivable that the dynamics can be learned more accurately through other means (by using prior knowledge, or by collecting more data with random exploration, etc.), and thus these experiments also suggest that our methods have a potential to perform better if the dynamics can be more accurate.
>
>
> Q11: “I think the name of the algorithm should be modified as "value iteration" refers to a specific dynamic programming algorithm for learning value functions, while the proposed algorithm does not resemble this at all. “
>
> Our algorithm VINS does learn value functions with iterative Bellman equation updates. We note that in line 4 of Algorithm 2, when we minimize the $\mathcal{L}_{\textrm{td}}$ part of the loss, we minimize the Bellman error. In our view, this can be viewed as an approximate dynamic programming. (But we are also happy to consider other names if the reviewer has some suggestions.)
>
> Minor comments:
> Thanks for pointing out the typos. We’ll fix them in the next version.
>
> For QT-Opt: Thanks for your kind suggestion. Yes, we agree that better optimization algorithms might result  in superior performance. We will mention this as a potential extension of our algorithm.
>
> [1] Overcoming Exploration in Reinforcement Learning with Demonstrations, Nair et al.
> [2] Hindsight Experience Replay, Andrychowicz et al.

---

### Decision · Program_Chairs · 2019-12-19

**Decision:**

Accept (Poster)

**Comment:**

The paper introduces Value Iteration with Negative Sampling (VINS) algorithm as a method to accelerate RL using expert demonstrations. VINS learns an initial value function that has a smaller value at states not encounter during the demonstrations.

The reviewers raised several issues regarding the assumptions, theoretical results, and experiments. The method seems to be most natural for robotic control problems. Nonetheless, it seems that the rebuttal addressed most of the concerns, and two of the reviewers increased their scores accordingly. Since we have three Weak Accepts, I believe this paper can be accepted at the conference.